# A Mediation Model between Self-Esteem, Anxiety, and Depression in Sport: The Role of Gender Differences in Speleologists

**DOI:** 10.3390/ijerph18168765

**Published:** 2021-08-19

**Authors:** Tamara de la Torre-Cruz, Isabel Luis-Rico, Cristina di Giusto-Valle, María-Camino Escolar-Llamazares, David Hortigüela-Alcalá, Carmen Palmero-Cámara, Alfredo Jiménez

**Affiliations:** 1Departamento de Ciencias de la Educación, Universidad de Burgos, 09001 Burgos, Spain; miluis@ubu.es (I.L.-R.); cdi@ubu.es (C.d.G.-V.); cpalmero@ubu.es (C.P.-C.); 2Departamento de Ciencias de la Salud, Universidad de Burgos, 09001 Burgos, Spain; 3Departamento de Didácticas Específicas, Universidad de Burgos, 09001 Burgos, Spain; dhortiguela@ubu.es; 4Department of Management, KEDGE Business School, 33405 Talence, France; alfredo.jimenez@kedgebs.com

**Keywords:** speleology, anxiety, depression, self-esteem

## Abstract

The scientific literature on mental health has found an association between physical activity and emotional wellbeing and recommends active leisure activities as a way of keeping stress under control. The purpose of this research study is to analyze the level of anxiety, the symptoms of depression and the level of self-esteem of people practicing speleology, as well as possible gender differences. This paper also attempts to understand whether self-esteem is associated with the presence of symptoms of depression in speleologists and whether anxiety has a mediating effect. We conduct a cross-sectional and descriptive research study with a sampling of 105 adult speleologists. The results reveal that the total mediation model is applicable, as self-esteem has a significant indirect association with depression through trait anxiety, as well as a partial mediation model that is applicable through state anxiety. This means that speleologists with high levels of self-esteem, who appreciate and value themselves adequately, reveal lower levels of trait anxiety, and this negatively influences their levels of depression (that is, a lower level of depressive symptoms). At the same time, speleologists with high levels of self-esteem, who appreciate and value themselves adequately, also reveal lower levels of state anxiety, which again has a negative impact on their levels of depression (with fewer symptoms of depression). Emotions such as anxiety, self-esteem, depression and their collateral effects are international topics of interest, which are relevant for people from all sporting backgrounds; therefore, value should be placed on supporting and carrying out further research into this topic.

## 1. Introduction

One of the major challenges for psychology in the 21st century is in extending its scope of action and studying the mechanisms that prevent poor health and promote good heath, with the purpose of promoting resources and quality of life in the population [1]. In this respect, interest in the study of motivation and adherence to physical exercise in relation to a global healthy lifestyle has been increasing [2,3]. The psychology applied to physical activity has been very productive since the end of the 20th century in the sense that instruments have been created and validated to help evaluate various cognitive and behavioral variables mediating the practice of physical exercise for health purposes [4].

The multifaceted and dynamic concept of wellbeing is closely associated with health, including subjective, social and psychological dimensions [1]. This concept directly relates physical/sports activities to the generation of positive emotional states that also relate to psychological wellbeing [5]. In this regard, physical activity presents itself as a special opportunity for the psychological flow state (optimal experience obtained without effort) to develop, as the practice of sport is related to many positive qualities and implies higher levels of involvement, desire, challenge and pleasure than other activities [6]. Specifically, the differences found between sports people and sedentary people suggest that physical activity reduces anxiety, depression and even stress [7,8,9].

Numerous research studies have highlighted the physical and psychological benefits that physical activity provides to humans [10,11,12]. In fact, regular and adapted physical exercise is one of the main strategies to promote active and healthy ageing, thus contributing to an improved quality of life [13,14].

Extreme sports imply participation in a pleasant and satisfactory activity [15]. Such sports share the common feature of entailing some actual or apparent danger to the physical integrity of its participants [16]. Extreme sports’ participants reveal a marked tendency to seek more intense experiences than people who practice less risky sports involving fewer types of experiences [17]. In fact, research has proven that individuals who score highly for the construct “seeking intense experiences” are interested in understanding the potential adverse consequences of their actions, but they always have a reduced perception of the risks and dangers involved in such activities in respect to the objective risks. Therefore, great intense-experience seekers share an “optimistic” and enterprising bias [18], thus perceiving lower risk and lesser probabilities of encountering negative consequences for their risky actions [19].

From a mental health perspective, valued and active sportive activities [20,21] are recommended as a negative emotions control practice [22]. An active lifestyle grants a protective effect against the incidence of disorders such as anxiety or depression. Previous studies have analyzed various popular sports, such as those by Aguinaga et al. [23], Hevilla-Merino and Castillo Rodríguez [24], and Muñoz Gómez et al. [25]; however, the literature has not covered other minority sports such as speleology, which can also play an important role in reducing the levels of depression and anxiety. This is why this research study focused on the practice of speleology as a sport. Given that it does not enjoy public funding, it is mainly practiced in leisure time, and thus can be considered active leisure. We also explore potential differences based on gender. The study sample was taken from a region in the North of Spain, given the large number of federate speleologists who practice this sport and the limited number of research studies on this sports group in Spain. 

### Anxiety, Depression and Self-Esteem in Sports

Despite the increased interest surrounding the role that emotions and psychological wellbeing play in the behavior of sportspeople, there are a number of situations and stimuli that the sportsperson can perceive as threatening [26], thus contributing to the creation of anxiety. According to Cheng et al. [27], anxiety in the context of sports is understood as an unwanted psychological state, as a reaction to the stress perceived as a result of performing a task under pressure [23,28].

On the one hand, anxiety has a cognitive component that includes concerns about performance, lack of concentration [29] and, therefore, inability to remain focused [30]. On the other hand, anxiety has a somatic component that is associated with the activation of physical symptoms such as an increase in heart rate [23,31,32].

It is also important to differentiate between trait anxiety and state anxiety. Trait anxiety refers to the predisposition to react anxiously towards a great variety of circumstances that may objectively not pose a physical or psychological threat [33]. On the other hand, state anxiety refers to the specific circumstances in which the sports person reacts anxiously, such as, for example, during the course of a competition [23,34,35].

The analysis of the differences in anxiety levels between sportsmen and sportswomen has proven to be controversial. For example, authors such as Ruiz-Juan et al. [36] do not find significant differences between the sexes in any dimension of anxiety in Spanish marathon runners. Other research finds higher averages for worry, deconcentration and global anxiety in the group of women compared to the group of men in samples of competitive athletes [37,38,39,40,41]. Borges et al. [42], in their study of adolescent water polo players, found higher scores in males for both precompetitive anxiety and competitive anxiety. The authors explain this finding based on the important maturational and psychobiological differences that exist between both sexes at puberty and adolescence.

On a different note, the symptoms of depression relate to a state of mind that can significantly interfere with the life of a human being in general, and particularly in sportspeople. The science has proven that when depression affects an individual, there is a significant alteration in the production of certain neurotransmitters that control their state of mind. According to scientific studies, in sports people specifically, the act of exercising their bodies helps to balance the production of neurotransmitters [25,43]. With regards to the gender of sportspeople, various studies have highlighted the protective effect of sport; that is, the more a sport is practiced, the fewer the symptoms of depression. In general, these studies all show that, for both men and women, it is not necessary to perform a high-volume physical activity, nor one that is vigorous in intensity. However, frequency is an important factor in reducing symptoms related to depression (the more frequent the physical activity, the lower the chances of depressive symptoms) [25,44]. Olmedillo et al. [45] found particular benefits in women who regularly practice some type of physical exercise.

In terms of self-esteem, this construct is considered a component or assessment version of the self-concept. This would, therefore, include cognitive, behavioral, affectionate and physical components [46]. The cognitive component relates to the global mental perception that an individual has of himself, and the affectionate component consists of a self-assessment made by the individual [47,48].

Prior research suggests that self-esteem is a relevant variable with regards to physical skills in sports performance [49]. Several studies relate self-esteem with self-performance [50], achievement and success, and with sports participation [47,51]. Rosenberg and Simmons [52] related self-esteem to psychological wellbeing and self-confidence, which lead to a consolidated personality and, therefore, result in the improvement in physical-sports performance [47,51]. In terms of the gender of the individual, the literature reveals that sportsmen have higher levels of self-esteem than sportswomen, which is potentially related to the gender gap—mostly universal—that can be found in different cultures [45,47]. This aspect is important, because, as Molina et al. [51] point out, self-esteem plays an important role in sports performance under conditions of high psychological pressure.

Based on the above, the purpose of this research paper is first to analyze the level of anxiety, the symptoms of depression and the self-esteem level of speleologists. Additionally, we attempted to analyze whether there are significant differences based on gender. Second, we aimed to understand whether self-esteem can cause symptoms of depression in speleologists, and whether (trait and state) anxiety mediates this association, with an impact on depression. 

## 2. Materials and Methods

### 2.1. Design

This paper is the result of the cross-sectional and descriptive study of a surveyed population, where the participants were selected through convenience sampling [53]. 

### 2.2. Participants

The sample consisted of 105 adult speleologists from the 632 (467 men and 165 women) members of the Spanish Federation of Speleology of Castilla y León (the Federation is the organization that promotes this physical activity and provides federate sportsmen and women with legal cover and training). The participants’ gender distribution reflects the natural distribution of the population of this sport, with 66 (62.9%) men and 39 (37.1%) women participating. Table 1 shows the characteristics of the participants, and highlights the distribution by age group, and the fact that the group of people between 31 and 45 years old, and the over-46 age group are homogeneous, and represent 80% of the sample. In terms of the level of education completed, over half of them have higher educational studies, with the groups who competed Vocational Training (VT), Primary and Secondary Studies being similar. In terms of the number of extreme sports they practice, the group practicing from 1 to 2 sports and the group practicing from 3 to 4 sports are homogeneous and represent approximately 80% of the sample. In terms of the distribution of the number of years they have been practicing speleology, the four groups are quite uniform, with approximately 25% of the sample falling in each of the ranges.

### 2.3. Instruments

A questionnaire was designed ad hoc in an online format, named an “Informative Questionnaire on Anxiety, Depression and Self-esteem” and structured in two differentiated parts. 

The first part consisted of the identification data, which comprised 11 questions dealing with aspects such as date of completion, age, autonomous region of residence, gender, nationality, marital status, level of studies completed, occupation and the number of years in their professional activity. Additionally, they were asked to mention the extreme sports they practice and whether they had taken advance training and/or risk prevention training courses, either for their professional activity or the sports activity in which they were involved. Prior to these questions, they were provided with a few quick instructions for the completion of the questionnaire and the basic ethical aspects of the research.

The second part consists of three inventories that are standardized and validated for a Spanish population: (a) State-Trait Anxiety Inventory (STAI) by Spielberger et al. [54], which was used to assess the state and trait anxiety levels of the participants, (b) Beck’s Depression Inventory (BDI) by Beck et al. [55], which was used to identify potential symptoms of depression, and (c) Rosenberg’s Self-esteem Scale (1965), which was used to assess the level of self-esteem.

The following are the main psychometric characteristics of the aforementioned data collection instruments.
-The State-Trait Anxiety Inventory (STAI) by Spielberger et al. [56] is a self-report consisting of 40 items, assessing two types of anxiety: State Anxiety (S-A: a temporary emotional condition) and Trait-Anxiety (T-A: a relatively-stable anxious tendency). These two subscales consist of 20 items within a 4-point Likert-type scoring system (where 0 = not at all, 1 = somewhat, 2 = quite, and 3 = A lot), which were designed to be administered individually or collectively. In Spanish population samples, the questionnaire had high indexes of internal consistency (0.90 to 0.93 for S-A and 0.84 to 0.87 for T-A) and high correlations with Cattell’s Anxiety Scale (0.73 to 0.85). The user manual for the test does not indicate the percentage at which anxiety is considered to be high, so the criteria used by Martínez-Otero [57], which consider the threshold for high anxiety to be percentages above 75 [58], was used.-The BDI test consists of 21 items that assess the intensity of depression. Each item consists of four statements, out of which the individual must chose the one that most reflects their personality and behavior. The options are in ascending order in terms of severity, which corresponds to a 0 to 3 Likert scale depending on the chosen option. The individual was assessed by considering their state of mind during the past week, including the day of the assessment. To draw the results from the test, the scores for each item were added to the depression score ranging between 0 and 63 points. As highlighted by Sanz et al. [59], the level of reliability of the BDI test’s internal consistency is high (alpha coefficient of 0.87). Factorial analyses indicated that the BDI test measures a general dimension of depression comprised of two highly related factors: one is cognitive-affective and the other is somatic-motivational. In terms of internal consistency and factorial validity, the BDI test appears to be one of the best instruments to assess symptoms of depression in the general population.-Rosenberg’s Self-Esteem Scale consists of 10 items, which are answered on a Likert-type scale (ranging from 1 = totally disagree to 4 = totally agree). It is comprised of five items that assess the level of positive self-esteem and another five assessing the level of negative self-esteem. The latter are re-codified with the purpose of obtaining a total self-esteem score. As Vázquez Morejón et al. [60] highlighted, there are several research studies that support its adequate psychometric characteristics in various languages [61].

### 2.4. Procedure

The questions relevant to the first part of the questionnaire (identification data) were content-validated by a group of experts. Following the analysis of the results obtained by the external jury, necessary changes were made to design the final questions. The questions allow us to contextualize participants within the object of study, establishing an alignment with the items under assessment in each of the scales. All the questions were sent by email through the Federation (The Federation of Speleology organizes and promotes this sport, as well as providing training. Being the holder of a federation license allows the members to be provided with insurance and cover while they perform their activities in the event of an accident, and offers discounts when entering certain events and/or purchase of sports equipment and access to grants, amongst other benefits) of Speleology of Castilla y León (Spain), which distributed it amongst the members of the various clubs. The email explained the purpose of the study, requested participation and included the link to the questionnaire, as well as giving the deadline for the application process. The completed questionnaires were received between September and November 2019, and data analysis was subsequently performed.

In terms of the code of conduct regulating the research, the participating sportsmen and women were informed, in writing (on the upper section of the first page of the questionnaire), of the voluntary and anonymous nature of their responses, as well as the fact that the data would only be used for research purposes. Therefore, if any of the sportsmen or women receiving the questionnaire were not in agreement with this, they could opt out.

### 2.5. Data Analysis

As previously mentioned, the purpose of this study is to analyze the anxiety, depression and self-esteem levels of of speleologists, compare them to the standard population, and explore potential gender differences. Subsequently, in order to study the differences with the standard population (subject to the relevant standardized benchmarks of age and gender) a Student’s *t*-test was performed with the summary answers from each sample. In order to compare the results of the sample, pertaining to anxiety, against the standard, the Spanish benchmark version of the STAI inventory [56] was used. The data on the level of depression were compared to the Spanish adaptation of the Beck Depression Inventory-II created by Sanz et al. [59] and, for the self-esteem questionnaire, the data from the adaptation and validation of Rosenberg’s Self-esteem scale produced by Gómez-Lugo et al. [62] were used. These data were processed in the statistical analysis software SPSS v.24 [63].

Finally, the mediation of the level of (trait and state) anxiety on the impact of self-esteem on depression was measured. In order to do this, a mediation analysis was employed through the bias-corrected and accelerated bootstrapping [64] method, with 10,000 bootstraps. For the mediation analysis, random bootstraps were performed on the main sample to estimate the direct effects (path c′) and indirect effects (path a and b and the total effect or path c) of the independent variable (self-esteem) and the mediating variable (anxiety) on the dependent variable (depression). This method was selected as the most adequate, because it is used on small samples, such as the one used in this research study, and does not require assumptions of normality and homoscedasticity. When the effect is significant, the model calculates the bootstrap’s confidence intervals (LLCI and ULCI). If the interval does not contain value 0, it is considered significant, and the mediation effect is considered to be present. These data were analyzed with the macro PROCESS for SPSS developed by Hayes [65]. Figure 1 shows a representation of the model under analysis. 

## 3. Results

To verify the first objective of the research, i.e., analyzing the level of anxiety, depression and self-esteem of speleologists, a Student’s *t*-test was performed on the summarized answers from each sample. The following are the most relevant findings.

### 3.1. Anxiety Scores

The scores for State Anxiety show differences that are statistically significant (*t* = −4.832; *df* = 355; *p* = 0.000) between the sample of male speleologists (*M* = 13.56; *SD* = 9.16) and the male standard sample (*n* = 295; *M* = 20.54; *SD* = 10.56), which reveals that male speleologists suffer lower levels of State Anxiety. In the case of female speleologists, there are differences that are statistically significant (*t* = −2.984; *df* = 400; *p* = 0.003) between the female speleologist sample (*M* = 17.16; *SD* = 11.84) and the male standard sample (*n* = 365; *M* = 23.30; *SD* = 11.93), which reveals that female speleologists also suffer from lower levels of State Anxiety (please refer to Table 2).

In terms of Trait Anxiety, there are differences that are statistically significant (*t* = −3.264; *df* = 378; *p* = 0.001) between the sample of male speleologists (*M* = 16.14; *SD* = 9.11) and the male standard sample (*n* = 318; *M* = 20.19; *SD* = 8.89), which implies that male speleologists reveal lower levels of Trait Anxiety. In terms of female speleologists, there are differences that are statistically significant (*t* = −3.010; *df* = 40.581; *p* = 0.004) between the sample of female speleologists (*M* = 18.62; *SD* = 12.48) and the female standard sample (*n* = 387; *M* = 24.99; *SD* = 10.05), which reveals that female speleologists also present lower levels of Trait Anxiety (please see Table 2).

### 3.2. Scores for Depression

In terms of the scores obtained for symptoms of depression in the general sample of speleologists, there are statistically significant differences (*t* = −3.313; *df* = 568; *p* = 0.001) between the speleologist sample (*M* = 6.58; *SD* = 7.86) and the standard sample (*M* = 9.40; *SD* = 7.70), which means that speleologists reveal generally lower levels of depression (please see Table 3).

In terms of the scores for symptoms of depression obtained by the male speleologist sample, there are statistically significant differences (*t* = −2.718; *df* = 282; *p* = 0.007) between the speleologist sample (*M* = 5.55; *SD* = 7.46) and the male standard sample (*n* = 233; *M* = 8.50; *SD* = 7.50), which shows that male speleologists reveal lower levels of depression (please see Table 3).

In the scores obtained for the sample of female speleologists, no statistically significant differences are found (*t* = −1.506; *df* = 284; *p* = 0.133) between the sample of female speleologists (*M* = 8.18; *SD* = 8.30) and the female standard sample (*n* = 247; *M* = 10.20; *SD* = 7.70), which reveals that female speleologists have the same levels of depression as the individuals in the standard sample (please see Table 3).

### 3.3. Scores for Self-Esteem

The results prove that there are no statistically significant differences regarding age or gender (*p* > 0.05) between the speleologist groups and the standard sample in terms of self-esteem, except for self-esteem in women over 45 years old, where female speleologists reveal lower levels of self-esteem than standard women within the same age range (please see Table 4).

Differences according to gender within the speleologists sample.

When analyzing the response according to gender, there are gender-based differences only in depression (*p* = 0.044) according to the depression scale, with the levels in women being significantly higher than levels in men. In terms of state anxiety, trait anxiety and self-esteem, there were no statistically significant differences in terms of gender (please see Table 5).

Differences according to age range within the speleologists sample, with age ranges of from 18 to 30; 31 to 45; and 45+.

The results show that there are no statistically significant differences according to the age range of speleologists (*p* > 0.05) for (state and trait) anxiety, depression or self-esteem (please see Table 6).

Differences according to years in the practice of speleology.

The results show that there are no statistically significant differences based on the number of years that speleology has been practiced (*p* > 0.05) for any of the variables under study (please see Table 7).

To compare the second objective, we aimed to determine whether the effects of self-esteem on depression were mediated by the levels of (trait and state) anxiety. Table 8 shows the results obtained in terms of the effects of self-esteem on depression with the mediation of (trait and state) anxiety. The results show that the total mediation model applies, and there is a significant indirect effect of self-esteem on depression through trait anxiety (*β* = −0.5069, BCa CI (−0.6717, −0.3080) and partial mediation through state anxiety (*β* = −0.3806, BCa CI (−0.5141, −0.2326) (please see Figure 2). Therefore, speleologists with high levels of self-esteem, who adequately appreciate and value themselves, reveal lower levels of trait anxiety, which positively impacts their depression levels (fewer symptoms of depression). In the same way, speleologists with high levels of self-esteem, who adequately appreciate and value themselves, also show lower levels of state anxiety, which negatively affects their depression levels (that is, fewer symptoms of depression).

## 4. Discussion and Conclusions

The first objective of this research study was to analyze the level of anxiety, the symptoms of depression and the level of self-esteem in people who practice speleology, as well as potential differences based on gender.

In terms of anxiety, both male and female speleologists reveal significantly lower levels of state and trait anxiety than the general population. These results are supported by other research studies with sportsmen and women carried out by Aguinaga et al. [23], Hevilla-Merino and Castillo Rodríguez [24], and Salom Martorell et al. [41]. Aguinaga et al. [23], in a research study with professional football players, proved that, amongst other factors, the protective role of group cohesion offered by this type of sport provided a reduction in the levels of competitive anxiety and an improvement in psychological wellbeing. Hevilla-Merino and Castillo Rodríguez [24], in a study with professional football players, saw a decrease in the levels of anxiety thanks to aspects such as the concentration of the player, motivation and attention involved in the practice of this sport. Salom Martorell et al. [41] also found a lower probability of experiencing high levels of anxiety in competitive sailors when they were involved in non-conditional, high sportive cooperation. These sportsmen and women enjoy a more pleasant and relaxed sporting experience.

This would mean that the practice of speleology as a sport could be seen to have a protective effect on emotional problems such as anxiety, a theory that is also supported by authors such as Muñoz Gómez et al. [25] in their research study with BMX sportsmen and women. Physical activity, in general, decreases the states of anxiety of people who practice them, due to the production of chemical substances that create wellbeing in the sportsmen or -women’s bodies. In much the same way, Cantón et al. [5] stated that experience in sports competitions plays a major role in the regulation of anxiety, as well as having an impact on psychological wellbeing in general.

In terms of symptoms of depression, we found that male speleologists suffer from significantly fewer symptoms of depression than the general male population. The results are not the same for women, for whom the differences are not significant. That is, female speleologists and women in the general population reveal similar levels of depression. These results are in contrast with the literature, which generally establishes an association between the practice of physical activity and a decrease in the levels of depression [45], in both men and women. That is to say, physical activity appears to have a protective effect in preventing individuals from suffering from depression or reducing the symptoms of depression, as Almagro Valverde et al. [44] stated. The results obtained for women in this research study can be related to the fact that, in the general population, women reflect higher levels of depression than men [66].

In terms of the levels of self-esteem, our research study does not find significant differences between men and women in the practice of speleology and the general population. The only exception is seen in female speleologists over 45, who clearly show lower levels of self-esteem. The lack of differences in the levels of self-esteem found between men and women is in contrast with the literature, which generally shows that people who practice physical exercise and sports, both men and women, tend to have higher levels of self-esteem and social self-concept than people who do not practice sports [67,68]. For women over 45, the results obtained in this research study are in line with the findings of other authors [45,68], where they state that the 45 to 54 age range is a vulnerable period of time for women and tends to show less self-esteem and more symptoms of depression [45]. Abalde Amoedo and Pino Juste [47] also saw, in their study of judo sportsmen and women, that men had higher levels of self-esteem than women. That is, they were able to see differences in the averages between men and women in terms of self-esteem and self-efficacy, confirming that sportsmen maintain higher levels of both self-perceptions than women. Their results suggest that judo sportsmen and women who have a higher level of perceived self-efficacy for their sports activity also attain a better sports performance.

A second objective posed by this research study was to understand whether self-esteem can cause symptoms of depression in speleologists in general, and whether this association is mediated by (state-trait) anxiety, also impacting depression. In this case, the potential mediating role of anxiety in the association between self-esteem and depression was analyzed, as this association was unlikely to be direct, and likely to be mediated by the anxiety levels of speleologists. The results for mediation indicate that trait and state anxiety effectively have a mediating role. Specifically, self-esteem influences depression in speleologists through anxiety. Therefore, speleologists with higher levels of self-esteem have lower levels of trait and state anxiety and this negatively influences the levels of depression (namely, lowers levels of depression), regardless of gender.

These results support the importance of practicing a sport such as speleology, since it seems to help maintain low levels of anxiety at both the trait level and state level.

These low levels of anxiety, in turn, will mediate the relationship between self-esteem and depression, favoring low depressive symptoms. In other words, anxiety is a key emotion in the well-being of athletes whose achieve control with regular sports practice. These results are in line with the research, such as: that of Ruiz et al. [69], with British athletes from a variety of team and individual sports; that of Stockbower et al. [70], with high school athletes; that of Kang and Jang [71], with Taekwondo athletes.

On the other hand, speleology, in addition to being a sport, is a form of research and exploration. Athletes who practice speleology show a passion for searching for the unknown, and a constant level of attention, which is required to move in a hostile environment. The stressful conditions to which the body is subjected and the ability to cope with them are acquired through experience [72,73]. Consequently, these variables also play a role in the emotions manifested by these types of athletes, such as self-esteem, depression, and anxiety.

This research study could potentially contribute to the field of emotions in sports practice, confirming the differences between groups of sportsmen and -women, although the direction of the differences differs, in some cases, from the differences found in other research studies, such as that of Guillén García and Niere Romero [68] with surfers, or that of Guillén and Álvarez-Malé [74] with competitive swimmers. Likewise, this study highlights the practice of sport in general, and speleology in particular, to improve self-esteem and combat depression. It also goes a step further, by distinguishing between trait and state anxiety as a mediating variable between self-esteem and depression in sports profiles.

In terms of the limitations of this research study, the following can be highlighted: the reduced size of the sample, the fact that the study focused on only one geographical area and the fact that it is a cross-sectional study. In terms of future research, studies that could shed new light on the influence of the practice of extreme sports on variables associated with emotional health would be of great interest. Future studies should also continue to investigate the differences between such groups, and further analyze the differences between specific sports and/or considering age groups, years of practice or intensity of practice, as some authors have already proposed [75]. By way of conclusion, we consider emotions such as anxiety, self-esteem, depression and their collateral effects to be interesting topics of international relevance, with significant repercussions in terms of the various sports [25]; therefore, these emotions demand further research.

## Figures and Tables

**Figure 1 ijerph-18-08765-f001:**
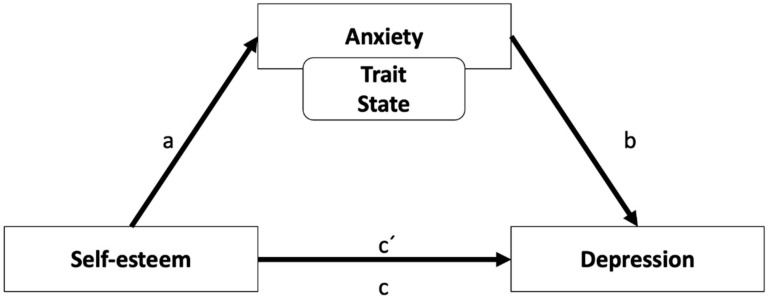
Mediation model under analysis. Note: a, b, c′ = direct effects; c = total effect.

**Figure 2 ijerph-18-08765-f002:**
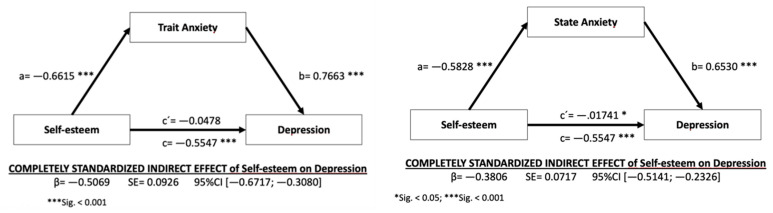
Mediation models under analysis. Note: a, b, c′ = direct effects; c = total effect.

**Table 1 ijerph-18-08765-t001:** Sociodemographic characteristics of the participants.

	Men	Women	Men	Women	Total	Total
Age						
18 to 30	13	8	61.9%	38.1%	21	100.0%
31 to 45	22	21	51.2%	48.8%	43	100.0%
over 46	31	10	75.6%	24.4%	41	100.0%
Total	66	39	62.9%	37.1%	105	100.0%
Education level completed						
Primary and secondary	16	7	69.6%	30.4%	23	100.0%
VT	21	4	84.0%	16.0%	25	100.0%
Upper	28	27	50.9%	49.1%	55	100.0%
Total	65	38	63.1%	36.9%	103	100.0%
No. of extreme sports *						
1 to 2	26	16	61.9%	38.1%	42	100.0%
3 to 4	27	16	62.8%	37.2%	43	100.0%
5+	13	7	65.0%	35.0%	20	100.0%
Total	66	39	62.9%	37.1%	105	100.0%
Years: speleology						
Less than 10	13	15	46.4%	53.6%	28	100.0%
11 to 20	16	9	64.0%	36.0%	25	100.0%
21 to 30	19	5	79.2%	20.8%	24	100.0%
31+	18	6	75.0%	25.0%	24	100.0%
Total	66	35	65.3%	34.7%	101	100.0%

* Extreme sports: canyoning; hiking; cave diving; snowboarding; rock climbing or climbing artificial rock walls; ski walking or snowshoeing; alpinism; canoeing/kayaking; recreational diving; BASE jumping; others.

**Table 2 ijerph-18-08765-t002:** State anxiety and trait anxiety values for the sample of speleologists and the standard sample.

State Anxiety		*M* ± *SD*	*t*	Degrees of Freedom	*p*
Male	Speleologists	13.56 ± 9.16	−4.832	355	0.000
	Standard	20.54 ± 10.56			
Female	Speleologists	17.16 ± 11.84	−2.984	400	0.003
	Standard	23.30 ± 11.93			
Trait anxiety					
Male	Speleologists	16.14 ± 9.11	−3.264	378	0.001
	Standard	20.19 ± 8.89			
Female	Speleologists	18.62 ± 12.48	−3.010 *	40.581 *	0.004 *
	Standard	24.99 ± 10.05			

* Hartley test for equal variance: Sig. < 0.05.

**Table 3 ijerph-18-08765-t003:** Values for symptoms of depression in the sample of speleologists and the standard sample.

Depression		*M* ± *SD*	*t*	Degrees of Freedom	*p*
General	Speleologists	6.58 ± 7.86	−3.313	568	0.001
	Standard	9.40 ± 7.70			
Male	Speleologists	5.55 ± 7.46	−2.718	282	0.007
	Standard	8.50 ± 7.50			
Female	Speleologists	8.18 ± 8.30	−1.506	284	0.133
	Standard	10.20 ± 7.70			

**Table 4 ijerph-18-08765-t004:** Values for self-esteem in the sample of speleologists and the standard sample.

Self-Esteem		*M* ± *SD*	*t*	Degrees of Freedom	*p*
18- to 30-year-old males	Speleologists	34.50 ± 5.00	0.998	87	0.321
	Standard (n = 77)	32.75 ± 5.74			
31- to 44-year-old males	Speleologists	35.52 ± 4.05	1.895	96	0.061
	Standard (n = 77)	33.36 ± 4.78			
45+ year-old males	Speleologists	34.10 ± 3.71	0.689	69	0.493
	Standard (n = 41)	33.39 ± 4.66			
18- to 30-year-old females	Speleologists	32.75 ± 3.99	0.833	189	0.406
	Standard (n = 183)	31.09 ± 5.57			
31- to 44-year-old females	Speleologists	34.33 ± 5.47	1.941	91	0.055
	Standard (n = 72)	31.70 ± 5.47			
45+ year-old females	Speleologists	31.77 ± 4.73	−2.454	56	0.017
	Standard (n = 49)	35.38 ± 3.92			

**Table 5 ijerph-18-08765-t005:** Results for the differences in the sample of speleologists according to sex.

	Gender	*M* ± *SD*	U	*p*
State anxiety	Men	13.56 ± 9.16	969	0.198
	Women	17.16 ± 11.84		
Trait anxiety	Men	16.14 ± 9.11	1103	0.750
	Women	18.17 ± 8.30		
Depression	Men	5.55 ± 7.46	905	0.044
	Women	8.17 ± 8.30		
Self-esteem	Men	34.65 ± 4.07	1065.5	0.352
	Women	33.39 ± 5.02		

**Table 6 ijerph-18-08765-t006:** Results for the differences in the speleologists sample according to age.

	Age Range	*M* ± *SD*	H	*p*
State anxiety	18 to 30 years old	13.75 ± 10.04	0.264	0.877
	31 to 45 years old	15.34 ± 10.87		
	Over 45 years old	15.05 ± 10.11		
Trait anxiety	18 to 30 years old	16.00 ± 12.98	1.559	0.459
	31 to 45 years old	17.14 ± 10.70		
	Over 45 years old	17.55 ± 9.01		
Depression	18 to 30 years old	6.47 ± 11.70	2.848	0.241
	31 to 45 years old	6.60 ± 6.92		
	Over 45 years old	6.60 ± 6.30		
Self-esteem	18 to 30 years old	33.80 ± 4.59	3.276	0.194
	31 to 45 years old	34.92 ± 4.79		
	Over 45 years old	33.56 ± 4.03		

**Table 7 ijerph-18-08765-t007:** Results for the differences in the speleologists sample according to years of practice.

	Years of Sports Practice	*M* ± *SD*	H	*p*
State anxiety	less than 10 years	17.18 ± 11.34	3.658	0.301
	11 to 20 years	15.20 ± 9.89		
	21 to 30 years	11.61 ± 8.16		
	Over 30 years	13.82 ± 10.29		
Trait anxiety	less than 10 years	18.37 ± 12.23	2.514	0.473
	11 to 20 years	18.87 ± 11.89		
	21 to 30 years	14.52 ± 7.17		
	Over 30 years	14.73 ± 7.40		
Depression	less than 10 years	7.59 ± 8.32	2.358	0.501
	11 to 20 years	6.20 ± 10.67		
	21 to 30 years	5.00 ± 4.40		
	Over 30 years	5.95 ± 4.56		
Self-esteem	less than 10 years	33.11 ± 4.94	5.566	0.135
	11 to 20 years	34.16 ± 4.49		
	21 to 30 years	36.13 ± 3.34		
	Over 30 years	33.86 ± 4.57		

**Table 8 ijerph-18-08765-t008:** Results of the mediation of self-esteem on depression through anxiety. * *p* < 0.05; *** *p* < 0.001.

Mediating Element	Effect of X (Self-Esteem) on M (Anxiety) (a)	*SD*	Effect of M on Y (Depression) (b)	*SD*	Effect of X on Y (c′)	*SD*	Estimated Bootstrap	*SD*	LLCI	ULCI
Trait	−0.6615 ***	0.1808	0.7663 ***	0.0633	−0.0478	0.1471	−0.5069	0.0926	−0.6717	−0.3080
State	−0.5828 ***	0.1943	0.6530 ***	0.0627	−0.1741 *	0.1446	−0.3806	0.0717	−0.5141	−0.2326

## Data Availability

The data presented in this study are available on request from the corresponding author. The data are not publicly available due to privacy issues.

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
