# Peer review of "A Mediation Model between Self-Esteem, Anxiety, and Depression in Sport: The Role of Gender Differences in Speleologists"

_ijerph, 2021, doi:10.3390/ijerph18168765_

Round 1

Reviewer 1 Report

The authors have attended to each of the issues raised in the reviewer report. 

Author Response

The authors have attended to each of the issues raised in the reviewer report.

RESPONSE: Thank you very much, we are glad that you find we were able to addressed each of the issues raised and we are grateful for your constructive feedback.

Reviewer 2 Report

The authors need to improve the sections:

  1. The relationship between the speleology and group activities.
  2. Gender differences antecedents in the introduction since it is one of their main hypotheses.
  3. Better quality in their figure resolution.
  4. The relationship between their title, research aims and importance of their findings in the Discussion section.
  5. The significance of their work, considering the relationship between trait and state anxiety and other variables and its meaning for speleology and sport's personality profiles.

And most importantly, to clarify the relationship (mediation) between their variables: sometimes it is not clear which one mediates.

Author Response

The authors need to improve the sections:

  1. The relationship between the speleology and group activities.
  2. Gender differences antecedents in the introduction since it is one of their main hypotheses.
  3. Better quality in their figure resolution.
  4. The relationship between their title, research aims and importance of their findings in the Discussion section.
  5. The significance of their work, considering the relationship between trait and state anxiety and other variables and its meaning for speleology and sport's personality profiles.

And most importantly, to clarify the relationship (mediation) between their variables: sometimes it is not clear which one mediates.

RESPONSE: Thank you very much for your suggestions, which have undoubtedly allowed us to improve the paper significantly. Next, we respond to each of the points that you have pointed out:

  1. In order to improve the relationship between the speleology and group activities, and also building on the feedback from Reviewer 3, we have reinforced the benefits that speleology can provide as a group sport. Besides, we have also removed from the discussion a sentence that wrongly gave the impression of speleology as an individual sport.

  1. In this revised version of the manuscript we have also emphasized more the antecedents of gender differences in the introduction, and we have reinforced this as one of our main aims in this research.

  1. Following your suggestion, we have improved the quality of the figures.

  1. Also following your feedback, we have modified the title to reflect more adequately the research aims and findings. The new title now reads as follows: A mediation model between self-esteem, anxiety, and depression in sport: the role of gender differences in speleologists. Further, we have tried to emphasize the importance of the findings as per your recommendations placed directly on the paper.

  1. We have taken into account your suggestion regarding the relationship between trait and state anxiety and other variables and its meaning for speleology and sport's personality profiles, trying to emphasize these aspects in the discussion of the manuscript.

Finally, regarding your last comment, we have changed several parts and corrected some typos in the paper that led to this confusion. Therefore, the mediation situation that we have found indicates that self-esteem has a significant indirect association with depression through trait anxiety; as well as a partial mediation model being applicable through state anxiety.

We apologize for this issues and thank you for your constructive feedback!

Reviewer 3 Report

Interesting article, but in my opinion it lacks a fundamental element: speleology is not only extreme sport. I don't know how it’s seen in Spain, but in Italy speleology is also (and perhaps above all) research and exploration. The passion for searching for the unknown, the constant attention required to move in a hostile environment, the stressful conditions to which the body is subjected and the ability acquired through experience to deal with them in the best possible way, certainly also play a role in self-esteem, anxiety and depression. It is not clear from the article whether these things have been considered. 

Another observation concerns what is said in paragraphs 459-461 about the individuality of speleology: it is not an individual activity, it’s always made in group (for safety reasons in the first place). The cohesion of the group is important in speleology and allows to face complex situations not only from a physical point of view, but also for the psychological support that others can offer.

Author Response

Interesting article, but in my opinion it lacks a fundamental element: speleology is not only extreme sport. I don't know how it’s seen in Spain, but in Italy speleology is also (and perhaps above all) research and exploration. The passion for searching for the unknown, the constant attention required to move in a hostile environment, the stressful conditions to which the body is subjected and the ability acquired through experience to deal with them in the best possible way, certainly also play a role in self-esteem, anxiety and depression. It is not clear from the article whether these things have been considered. 

RESPONSE: We are very glad to find that you think our article is interesting and we thank you very much for your feedback. We totally agree with you regarding the specific characteristics of speleology and we now reflect them in the paper.

Another observation concerns what is said in paragraphs 459-461 about the individuality of speleology: it is not an individual activity, it’s always made in group (for safety reasons in the first place). The cohesion of the group is important in speleology and allows to face complex situations not only from a physical point of view, but also for the psychological support that others can offer.

RESPONSE: Thank you very much again. You are right, this sentence is misleading and not correct. In this revised version of the manuscript we have removed this sentence from the discussion section and, instead, we have reinforced the benefits of speleology as a group activity.

This manuscript is a resubmission of an earlier submission. The following is a list of the peer review reports and author responses from that submission.

Round 1

Reviewer 1 Report

In general, the contribution to the knowledge base in this domain (wellbeing and sports participation) is not of sufficiently interesting quality to warrant publication. Moreover, the quality of the language detracts from the paper doing the paper an even greater disservice. The findings are not unexpected and the various T test results are predicable. In sum, there is nothing new to be learned from this study.

Table 1:

No. of extreme sports  categories should read as

1-2

3-4 (not 2-4)

5+

Results: it is stated that the male and female sub-samples are compared to the ‘standard sample’ for each scale. The assumption is that this standard sample consists of both male and female respondents, but it is not made clear in the narrative (it is made clear in Table 2).

Degrees of freedom is typically symbolized using “df” in English and not “gl”.

There is a lack of consistency in decimal reporting (in narrative vs tabular format). Rather report up to two decimals throughout.

It would be informative to include the sample size for the standard sample when comparing the two samples for both genders.

There is no need to include both the narrative and tabular output of the same data. The narrative results for the various T tests appear again in the tables. Note that in Table 2 there is an error for the male standard score for the trait anxiety result (0.19 which should read 20.19).

The degrees of freedom for the female sample for the trait anxiety scores looks incorrect (when compared to the other results) with a figure of 40.58. The rest of the df figures look odd – are you sure they are correct?

English grammar can be improved and there are several stylistic features which detract from this paper. It is advised that a native English first language speaker proofread the document.

The column headings in Table 8 should ideally use the wording in-text and not use abstract labels such as ‘X on Y’. Figure 2 headings must be translated into English. In figure 2, the first mediation diagram is a repetition of the one below it. This is simply a proofing issue.

What statistical software programme was used to analyse the data? It is customary to state this. The discussion around the medication is very sparse.

Reviewer 2 Report

Dear authors,

The study entitled "Effects of the Practice of Speleology on Emotional Wellbeing: Anxiety, Depression and Self-Esteem" is a cross-sectional study, composed of a small convenience sample (105 subjects). 
As a strong point of the study, it is probably the first descriptive study of these characteristics carried out with Spanish speleologists.
Despite this, the title of the study does not agree with the method used or with its objectives, as the action of the exercise (frequency, duration, intensity, etc.) on the variables measured is not assessed at any point. To be able to do this, the study should have been longitudinal, with at least one control group that did not go caving for a sufficient period of time to demonstrate the cause-effect relationship.
With a cross-sectional study, cause-effect relationships cannot be established.
In addition to all this, the justification for their study (lines 66-69) is unrelated to what they are investigating using the method described. At no point do you measure the variables exercise type, frequency, duration, intensity. You have measured state/trait anxiety, self-esteem, socio-demographic variables.

You indicate that with these descriptive and cross-sectional data you have carried out a comparison of means with Student's test. 
From a descriptive point of view, it may make sense to compare the means of quantitative variables in relation to sex, etc., but with the limitations of the method it is not possible to use these means to establish cause-effect relationships.
Furthermore, they carry out a mediation analysis to find out the cause-effect relationship between the constructs with cross-sectional data, which is impossible to do for the reasons I have argued above.

I recommend that before applying analyses to data, you try to understand the requirements of the analyses you are applying. Scientific methodology has methodological bases that must be known before applying the analyses.

Your study would have been an interesting descriptive study if you had not incurred in all these methodological flaws. I hope that my comments will be useful for you to learn and improve your work.

Best regards,